# Prenatal Diagnosis and Fetopsy Validation of Complete Atrioventricular Septal Defects Using the Fetal Intelligent Navigation Echocardiography Method

**DOI:** 10.3390/diagnostics13030456

**Published:** 2023-01-26

**Authors:** Paola Veronese, Alvise Guariento, Claudia Cattapan, Marny Fedrigo, Maria Teresa Gervasi, Annalisa Angelini, Arianna Riva, Vladimiro Vida

**Affiliations:** 1Maternal-Fetal Medicine Unit, Department of Women’s and Children’s Health, University of Padua, 35128 Padova, Italy; 2Pediatric and Congenital Cardiac Surgery Unit, Department of Cardiac, Thoracic, Vascular Sciences and Public Health, University of Padua, 35128 Padova, Italy; 3Cardiovascular Pathology Unit, Department of Cardiac, Thoracic and Vascular Sciences and Public Health, University of Padua, 35128 Padova, Italy

**Keywords:** atrioventricular septal defects, congenital heart disease, spatiotemporal image correlation, fetal intelligent navigation echocardiography, virtual intelligent sonographer

## Abstract

(1) Background: Artificial Intelligence (AI) is a modern tool with numerous applications in the medical field. The case series reported here aimed to investigate the diagnostic performance of the fetal intelligent navigation echocardiography (FINE) method applied for the first time in the prenatal identification of atrioventricular septal defects (AVSD). This congenital heart disease (CHD) is associated with extracardiac anomalies and chromosomal abnormalities. Therefore, an early diagnosis is essential to advise parents and make adequate treatment decisions. (2) Methods: Four fetuses diagnosed with AVSD via two-dimensional (2D) ultrasound examination in the second trimester were enrolled. In all cases, the parents chose to terminate the pregnancy. Since the diagnosis of AVSD with 2D ultrasound may be missed, one or more four-dimensional (4D) spatiotemporal image correlation (STIC) volume datasets were obtained from a four-chamber view. The manual navigation enabled by the software is time-consuming and highly operator-dependent. (3) Results: FINE was applied to these volumes and nine standard fetal echocardiographic views were generated and optimized automatically, using the assistance of the virtual intelligent sonographer (VIS). Here, 100% of the four-chamber views, and after the VISA System application the five-chamber views, of the diagnostic plane showed the atrioventricular septal defect and a common AV valve. The autopsies of the fetuses confirmed the ultrasound results. (4) Conclusions: By applying intelligent navigation technology to the STIC volume datasets, 100% of the AVSD diagnoses were detected.

## 1. Introduction

Atrioventricular septal defects (AVSDs) are found in congenital heart disease (CHD), characterized by a common atrioventricular junction and a central septum deficiency resulting from the fusion failure of the endocardiac cushions during the embryogenesis of the heart [1,2]. AVSDs are found in 4–5% of all infants with CHD and occur in 0.19 per 1000 live births [3,4]. Nowadays the prenatal diagnosis of CHD, including AVSDs, is performed using two-dimensional (2D), three-dimensional (3D), or four-dimensional (4D) fetal echocardiography [5,6,7,8,9,10,11,12,13,14,15].

Volumetric sonography, and specifically four-dimensional (4D) ultrasound with spatiotemporal image correlation (STIC), facilitates the examination of the fetal heart, and has been proposed in both the cardiac screening and prenatal diagnosis of CHD. Indeed, this method allows the identification of complex intracardiac relationships with better accuracy and shortens the examination time.

STIC technology enables the acquisition of a volume dataset of the fetal heart, displaying a cine loop of a single cardiac cycle in motion. Therefore, such technology has made it feasible to theoretically capture all information (i.e., fetal cardiac anatomy) incorporated within the transducer sweep. Sonologists can then interrogate the volume dataset and examine anatomical areas of interest in planes of a section other than in the original acquisition plane. Unfortunately, retrieving and displaying all of the relevant cardiac views with this “manual navigation” method (e.g., operating the x, y, z controls, scaling, parallel shifting) is difficult, time-consuming, operator-dependent, and requires a deep understanding of the cardiac anatomy.

In this field, algorithms based on STIC have been developed over the years to get over the operator’s dependency and manual manipulation by exploiting the AI. The fetal intelligent navigation echocardiography (FINE) method represents a step forward compared to other techniques for fetal cardiac views [16,17,18]. This method allows the automatic generation and display of nine standard fetal echocardiography views in normal hearts, including those universally recommended by professional organizations, such as the American Institute of Ultrasound in Medicine (AIUM) and International Society of Ultrasound in Obstetrics and Gynecology (ISUOG) [19,20].

Virtual Intelligent Sonographer Assistance (VIS-Assistance®) can be used together with FINE as an operator-independent sonographic tool to navigate and explore the surrounding structures in each of the nine cardiac diagnostic planes in order to reduce the false-positive rate and improve the quality of the examination [16,17,18].

Acquiring an STIC volume is a relatively simple procedure that depends purely on the position of the fetus. Recently, it has been reported that in women between 19 and 30 weeks of gestation with a normal fetal heart, STIC volumes can be successfully obtained in 72.5% of cases. Furthermore, it has been shown that FINE can be applied in 98–100% of cases using a combination of diagnostic planes or VIS-Assistance [16].

The aim of this study was to test the FINE method for prenatal diagnosis in a series of fetal AVSD patients. After marking seven anatomical structures of the fetal heart, the nine standard fetal echocardiography views were automatically generated and displayed using FINE. The diagnosis of AVSD was confirmed at the time of fetal autopsy after the termination of the pregnancy.

## 2. Materials and Methods

This was a prospective case report series of patients with a prenatal diagnosis of AVSD at the Unit of Maternal Fetal Medicine, Department of Women and Children’s Health, University of Padua, Italy, with subsequent interruption of the pregnancy. All patients were enrolled in a research protocol approved by the local ethics committee and the parents provided their written informed consent for the use of ultrasound images for research purposes. The hospital has a Federalwide Assurance (FWA) agreement negotiated with the Office for Human Research Protections (OHRP), U.S. Department of Health and Human Services.

The patients underwent a 2D sonographic examination and 4D sonography with STIC between November 2016 and July 2020. Using STIC technology (Voluson E8 Expert; GE Healthcare, Zipf, Austria), 4D volume datasets of the fetal heart (1–5 per patient) were obtained from the apical four-chamber view via transverse sweeps through the fetal chest. The patient was asked to momentarily suspend their breathing during the STIC volume acquisition; the acquisition time was 12.5 s. Attempts were made to acquire STIC volume datasets when the fetal spine was located between the 5 and 7 o’clock positions, to reduce the likelihood of acoustic shadowing derived from the spine.

The STIC volumes were acquired. All STIC volume datasets were saved and then imported into a software system for the analysis (SONOCUBIC FINE™ Classic Blue Series; Version 2013.04.05; Medge Platforms Inc., New York, NY, USA), which was installed on a SONY VAIO PCG 71211M desktop computer (SONY Corp., Minato, Tokyo, Japan) using the Microsoft Windows 7 PRO 64-bit service pack (Microsoft Corp., Redmond, WA, USA). Once an STIC volume was loaded into this software, it was immediately converted into a two-dimensional cine loop that automatically scrolled in a continuous fashion (i.e., STICLoop™) [16]. This tool allowed the sonologist to determine the appropriateness of the STIC volume datasets before applying the FINE method [16,17,18].

FINE was then applied to these volumes to analyze the STIC dataset stored in the DICOM files. After marking seven anatomical structures of the fetal heart, nine standard fetal echocardiography views were automatically generated and displayed using FINE (as diagnostic planes or VIS-Assistance®).

Details of the pregnancy and neonatal outcome, autopsy results, and karyotype analysis were recorded. VIS-Assistance diagnostic plans and video clips were selected for the presentation.

## 3. Results

During the study period, 4 cases were recruited (Table 1):Case 1: A 30-year-old woman (G2, P1001) referred to our maternal fetal unit at 20 weeks of gestation. She had received amniocentesis at 16 weeks of gestation after a combined test at high risk for Down’s syndrome. The fetal karyotype result was 47 XY + 21. The pregnancy had arisen spontaneously.Case 2: A 35-year-old woman (G2, P1001) referred to our maternal fetal unit at 20 weeks of gestation. She had received amniocentesis at 16 weeks of gestation due to maternal choice. The fetal karyotype result was 47 XX + 21. The pregnancy had arisen spontaneously.Case 3: A 28-year-old woman (G1, P 0000) referred to our unit at 18 weeks of gestation. She had received chorionic villus sampling at 13 weeks of gestation after a first-trimester scan showed a fetal cystic hygroma. The fetal karyotype result was 47 XX + 21. The pregnancy had arisen spontaneously.Case 4: A 38-year-old woman (G3, P 0020) referred to our unit at 17 weeks of gestation. She had received chorionic villus sampling at 12 weeks of gestation due to her maternal age. The fetal karyotype result was 47 XX + 21. The pregnancy was obtained with the FIVET procedure with a homologous biological material.

All cases were referred to our unit due to a prenatal diagnosis of trisomy 21 and to receive a prenatal ultrasound. Complete AVSDs were observed via 2D sonography and confirmed in all cases. No other malformations were observed.

One or more STIC volumes (totally 20) were acquired. From the STIC volumes determined to be appropriate using STICLoop™, only a single volume per patient was selected for the analysis using the FINE method. When there were multiple appropriate STIC volumes available per fetus, we chose the dataset of highest quality. Nine cardiac diagnostic planes in a single template with the additional feature of automatic labeling through intelligent navigation were extracted (Figure 1a,b).

The STIC volume was acquired and analyzed using the FINE method, which showed an atrial septal primum defect and a ventricular septal defect in the four-chamber view. At that time, we also detected the abnormal common atrioventricular valve, but in none of the 4 cases was a ventricular disproportion found. With the VISA application, the 5-chamber view showed the AVSD defect and 4 separate pulmonary veins in the left atria. All other diagnostic planes appeared to be normal (Table 2)

All patients, after multidisciplinary counseling with a geneticist, pediatric cardiologist, and obstetrician expert in maternal fetal medicine, elected for termination of pregnancy (TOP). The TOP was carried out pharmacologically with the use of vaginal gemeprost. The autopsies of the fetuses confirmed the sonographic findings (Figure 2), excluding other cardiac and extra cardiac anomalies.

## 4. Discussion

The prenatal diagnosis of a complete AVSD is not always straightforward. Fetal AVSDs are usually identified by the echocardiographic hallmark of a common atrioventricular valve and the distortion of the normal appearance at the crux of the heart. Upon 2D ultrasound, when the atrial and septal defects are large, the four-chamber view reveals an obvious deficiency of the central core structures of the heart [21].

The prenatal diagnosis of AVSD has been reported using 2D, 3D, and 4D sonography methods with STIC [5,6,15]. The advantage of 4D and STIC is that they provide the possibility to diagnose complex malformations that may not be easily identified with traditional 2D imaging, thereby reducing the operator dependency. As a matter of fact, the use of 4D ultrasound with STIC and FINE has been shown to help the operator during detailed sonographic evaluations of the fetal heart in cases of CHD [6,9,10,11,12,13,14].

Several studies showed how these methods could be used in prenatal screening, helping with diagnosing CHD. Algorithms based on STIC have been developed to automate the retrieval of cardiac diagnostic planes and to reduce operator dependency. Recently, the FINE methodology was developed to interrogate an STIC volume dataset. By applying “intelligent navigation” technology to the STIC volume datasets, the nine standard fetal echocardiography views are automatically generated. Additionally, the new VIS-Assistance allows the spontaneous navigation of the surrounding anatomy in each of the nine cardiac diagnostic planes.

Romero et al. sought to determine the sensitivity and specificity of FINE in the prenatal detection of CHD and found that FINE identifies a large spectrum of CHD cases with 98% sensitivity and 93% specificity [22]. The CHD cases identified in this study were conotruncal anomalies, left and right heart anomalies, complex cardiac defects, and septal defects. Furthermore, these authors demonstrated how the FINE method can also be successfully applied in the screening (and subsequent diagnosis) of other CHD cases, such as tetralogy of Fallot with pulmonary atresia, hypoplastic left heart, coarctation of the aorta, and prenatal diagnosis of dextrocardia [23,24,25,26].

In addition to this, the FINE tool can also be applied to STIC volumes acquired with color. This offers the possibility of obtaining both anatomical and functional information, an option valid for both normal hearts and those with congenital heart disease [27]. The possibility of being able to use this tool reduces the subjectivity linked to the operator and greatly simplifies the fetal cardiac investigation, thanks to the manipulation and analysis of STIC volume datasets [16,17,18].

All of these concepts can be applied for the anatomical study of atrioventricular valves, both normal and pathological. In our study, a volume dataset of the fetal heart was acquired through the STIC function and a cine loop of a complete single cardiac cycle in motion was displayed. The cardiac planes can be extracted and presented in any orientation by analyzing three orthogonal planes in multiplanar display mode.

In these case series, 100% of the four-chamber views, and after the VISA System application the five-chamber views, of the diagnostic plane showed the atrioventricular septal defect and a common AV valve. This demonstrates how additional methods can facilitate the diagnosis of anomalies of the atrioventricular valves. In this sense, the case in which we successfully applied the FINE method to a fetus with a complete AVSD at 20 weeks of gestation was emblematic.

This is the first report recognizing the FINE method as a useful tool in the prenatal identification of AVSD. Many of the larger studies report an association range of 13–72% of AVSD with extracardiac anomalies. The range of extracardiac abnormalities is wide and includes renal, gastrointestinal, neurological, and skeletal abnormalities. An association range of 37–58% of AVSD, commonly of the complete type, with chromosomal anomalies, and especially with trisomy 21, has been described. Therefore, the fetal karyotype should be examined whenever this diagnosis is made [1,2,3,4].

The detection of associated abnormalities (both extracardiac and chromosomal) is very important to provide an accurate indication of outcomes when counseling parents. Indeed, the outcomes of fetal AVSD depend on associated cardiac and extracardiac lesions, with high incidence rates of pregnancy termination and neonatal death [28]. It appears that tool such as FINE can facilitate and improve the detection of this difficult cardiac anomaly and can identify these fetuses with frequent extracardiac and chromosomal alterations. It is also well recognized that the outcomes of CHD diagnosed prenatally differ from those cases detected after birth [29,30]. The more accurate system we can use to obtain the diagnosis, the better outcomes we can guarantee for our patients.

## Figures and Tables

**Figure 1 diagnostics-13-00456-f001:**
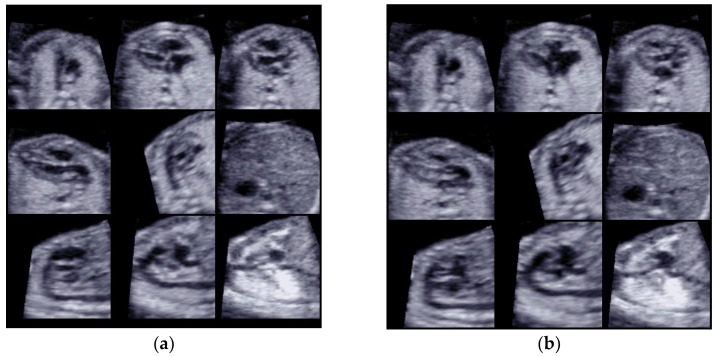
(**a**) Diagnostic planes with FINE in the systolic cardiac phase; 9 planes were used for the analysis. (**b**) Diagnostic planes with FINE in the diastolic cardiac phase; 9 planes were used for the analysis.

**Figure 2 diagnostics-13-00456-f002:**
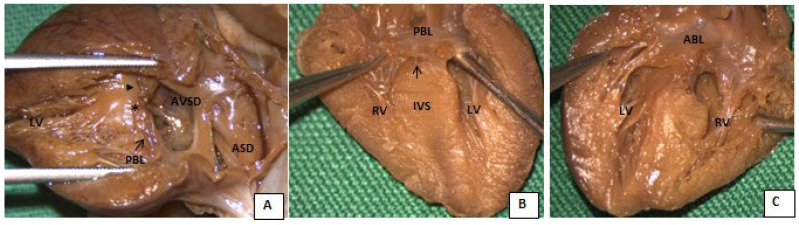
Female fetus heart at 20 weeks of gestation, affected by trisomy 21 and volunteer pregnancy interruption. (**A**) Macroscopic view of the left cavities mimicking the lateral long axis two-chamber echo view; note the patent oval fossa and complete atrioventricular septal defect with a common orifice of the valve with the posterior bridging leaflet (PBL, arrow) and anterior bridging leaflet (head arrow) and the gap between the two (*). Four-chamber section at the level of the gap at the interventricular septum between the anterior and posterior bridging leaflets. Posterior view (**B**) showing the posterior bridging leaflet attached to the interventricular septum (arrow), with the intact free margin of the posterior leaflet confirming the absence of the fusion with the anterior leaflet. Anterior view of the four-chamber section with the anterior leaflet attached to the interventricular septum (type A of Rastelli) (**C**).

**Table 1 diagnostics-13-00456-t001:** Clinical features and pregnancy outcomes of patients with fetal atrioventricular septal defects.

N	Maternal Age	Gestational Age at US	Cardiac Defect at 2D US	Chromosomal Abnormalities	Pregnancy Outcome	AVSD Confirmation
1	30	20	Complete AVSD	47 XY + 21	TOP	Autopsy: complete AVSD
2	35	20	Complete AVSD	47 XX + 21	TOP	Autopsy: complete AVSD
3	28	18	Complete AVSD	47 XX + 21	TOP	Autopsy: complete AVSD
4	38	17	Complete AVSD	47 XX + 21	TOP	Autopsy: complete AVSD

**Table 2 diagnostics-13-00456-t002:** Abnormal findings from the 2D echocardiography and after using FINE (both in the diagnostic planes and after VIS-Assistance) and the different findings during systolic and diastolic phases.

	2D echo	FINE	FINE	FINE	FINE	FINE
N	Abnormal Finding	Abnormal Diagnostic plane	Abnormal Finding	Abnormal VISA Views	Abnormal VISA Finding	Findings in Systolic/Diastolic Phase
1	Complete AVSD	4 CH	AVSD + AV valves in the same plane	4 CH 5 CH	AVSD + common AV valve AVSD + common AV valve	Normal/AVSD + common AV valve
2	Complete AVSD	4 CH	AVSD + Common AV valve	4 CH 5 CH	AVSD + Common AV valve AVSD + Common AV valve	AV valves in the same plane + AVSD/Common AV valve + AVSD
3	Complete AVSD	4 CH	AVSD + Common AV valve	4 CH 5 CH	AVSD + Common AV valve AVSD Common AV valve	AV valves in the same plane + AVSD/Common AV valve + AVSD
4	Complete AVSD	4 CH	AVSD + Common AV valve	4 CH 5 CH	AVSD + Common AV valve AVSD + Common AV valve	AV valves in the same plane + AVSD/Common AV valve + AVSD

## Data Availability

Data presented in this study are available on request from the corresponding author.

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
