# Peer review of "Prenatal Diagnosis and Fetopsy Validation of Complete Atrioventricular Septal Defects Using the Fetal Intelligent Navigation Echocardiography Method"

_diagnostics, 2023, doi:10.3390/diagnostics13030456_

Round 1

Reviewer 1 Report

1) Introduction L74-76.The aim of this study was to report a case series of AVSD patients for whom FINE was used for prenatal diagnosis with subsequent validation at fetal autopsy after termi- nation of pregnancy. Please improve the description of study aim.

2) Discussion. L 216-220. The detection of associated abnormalities (both extracardiac than chromosomal) is  very important to provide an accurate indication of outcomes when counseling parents.  Indeed, the outcomes of fetal AVSD depend on associated cardiac and extracardiac le-  sions, with a high incidence of pregnancy termination and neonatal death (28). It is well  recognized that the outcomes of CHD diagnosed prenatally differs from those detected  after birth [29,30]. Please, add a separate paragraph withthe conclusion of the study.

Author Response

  1. The aim of this study was to test the FINE method for prenatal diagnosis in a case series of fetal AVSD patients. After marking seven anatomical structures of the fetal heart, the nine standard fetal echocardiography views were automatically generated and displayed by FINE. The diagnosis of AVSD was confirmed at fetal autopsy after termination of pregnancy.
  2. In these case series, 100% of the four-chamber and, after the VISA System
    application, five-chamber view diagnostic plane showed the atrioventricular septal defect and a common AV valve. This demonstrates how additional methods can facilitate the diagnosis of anomalies of the atrioventricular valves. In this sense, emblematic is the case in which we successfully applied the FINE method to a fetus with a complete AVSD
    at 20 weeks of gestation.
    This is the first report recognizing the FINE method as a useful tool in the prenatal identification of AVSD. Many of the larger studies report an association 13–72% of AVSD with extracardiac anomalies. The range of extracardiac abnormalities is wide and includes renal, gastrointestinal, neurological, and skeletal abnormalities. An association of 37–58% of AVSD, commonly of the complete type, with chromosomal anomalies, and
    especially with trisomy 21, has been described. Therefore, the fetal karyotype should be examined whenever this diagnosis is made [1–4] .
    The detection of associated abnormalities (both extracardiac than chromosomal) is very important to provide an accurate indication of outcomes when counseling parents.
    Indeed, the outcomes of fetal AVSD depend on associated cardiac and extracardiac lesions, with a high incidence of pregnancy termination and neonatal death (28). It appears that a tool as FINE can facilitate and improve the detection of this difficult cardiac anomaly and identify these fetuses with frequent extracardiac and chromosomal alterations. It is also well recognized that the outcomes of CHD diagnosed prenatally differs from those detected after birth [29,30]. The more accurate system we could use to get the diagnosis, the better outcomes we can garantee to our patients.

Reviewer 2 Report

To:

Editorial Board

Diagnostics

Title: “Prenatal Diagnosis and Fetopsy Validation of Complete Atrioventricular Septal Defects Using the Fetal Intelligent Navigation Echocardiography Method”

Dear Editor,

I read this paper and I think that:

-          The small sample size is a limitation of the paper. Despite the relative rarity of the condition, the sample size is too small for a definite original article. Authors can convert the paper into a case report series.

-          The reproducibility of the echocardiographic methods should be validated and assessed. Inter and intraobserver variability coefficients should be calculated. Please provide.

-          Results section of the abstract should include more numerical data. Please update this section.

Author Response

  1. Dear reviewer,
    you are completely right, indeed we have already converted the paper in a case report series in agreement with the editor.
  2. Acquiring a STIC volume is a relatively simple procedure that depends purely on the position of the fetus. Recently, it has been reported that in women between 19-30 weeks of gestation with a normal fetal heart, STIC volumes can be successfully obtained in 72.5% of cases. Furthermore, it has been shown that FINE can be applied in 98–100% using a combination of diagnostic planes and/or VIS-Assistance, automatically generating the nine standard echocardiography view. The validation of fetal intelligent
    navigation echocardiographic method is described in articles 16 and 18.*Yeo L, Romero R. Fetal Intelligent Navigation Echocardiography (FINE): a novel method for rapid, simple, and automatic examination of the fetal heart. Ultrasound Obstet Gynecol Off J Int Soc Ultrasound Obstet Gynecol. 2013 Sep;42(3):268–84. **Veronese P, Bogana G, Cerutti A, Yeo L, Romero R, Gervasi MT. A Prospective Study of the Use of Fetal Intelligent Navigation Echocardiography (FINE) to Obtain Standard Fetal Echocardiography Views. Fetal Diagn Ther. 2017;41(2):89–99.
  3. Results: FINE was applied to these volumes and nine standard fetal echocardiographic
    views were generated and optimized automatically, using the assistance of the Virtual
    Intelligent Sonographer (VIS). 100% of the four-chamber and, after the VISA System application, five-chamber view diagnostic plane showed the atrioventricular septal defect and a common AV valve. Autopsies of the fetuses confirmed the ultrasound results.
